# NV-Embed: Improved Techniques for Training LLMs as Generalist Embedding Models

Chankyu Lee [*1]        Rajarshi Roy [1]        Mengyao Xu [1]        Jonathan Raiman [1]

Mohammad Shoeybi [1]        Bryan Catanzaro [1]        Wei Ping [*1]

**NVIDIA**

## Abstract

Decoder-only large language model (LLM)-based embedding models are beginning to outperform BERT or T5-based embedding models in general-purpose text embedding tasks, including dense vector-based retrieval. In this work, we introduce the `NV-Embed` model, incorporating architectural designs, training procedures, and curated datasets to significantly enhance the performance of LLM as a versatile embedding model, while maintaining its *simplicity* and *reproducibility*. For *model architecture*, we propose a *latent attention layer* to obtain pooled embeddings, which consistently improves retrieval and downstream task accuracy compared to mean pooling or using the last `<EOS>` token embedding from LLMs. To enhance representation learning, we remove the causal attention mask of LLMs during contrastive training. For *training algorithm*, we introduce a two-stage contrastive instruction-tuning method. It first applies contrastive training with instructions on retrieval datasets, utilizing in-batch negatives and curated hard negative examples. At stage-2, it blends various non-retrieval into instruction tuning, which not only enhances non-retrieval task accuracy but also improves retrieval performance. For *training data*, we utilize the hard-negative mining, synthetic data generation and existing public available datasets to boost the performance of embedding model. By combining these techniques, our NV-Embed-v1 and NV-Embed-v2 models obtained the No.1 position on the Massive Text Embedding Benchmark (MTEB) (as of May 24, 2024 and August 30, 2024, respectively) across 56 embedding tasks, demonstrating the sustained effectiveness of the proposed methods over time. Also, it achieved the highest scores in the Long Doc section and the second-highest scores in the QA section of the AIR Benchmark, which covers a range of out-of-domain information retrieval topics beyond those in MTEB. We further provide the analysis of model compression techniques for generalist embedding models. We open-source the model at: https://huggingface.co/nvidia/NV-Embed-v2.

## 1 Introduction

Embedding or dense vector representation of text (Mikolov et al., 2013; Devlin et al., 2018) encodes its semantic information and can be used for many downstream applications, including retrieval, reranking, classification, clustering, and semantic textual similarity tasks. The embedding-based retriever is also a critical component for retrieval-augmented generation (RAG) (Lewis et al., 2020), which allows LLMs to access the most up-to-date external or proprietary knowledge without modifying the model parameters (Liu et al., 2024; Guu et al., 2020; Shi et al., 2023; Wang et al., 2023a).

The embedding models built on bidirectional language models (Devlin et al., 2018; Raffel et al., 2020) have dominated the landscape for years (e.g., Reimers & Gurevych, 2019; Gao et al., 2021; Wang et al., 2022; Izacard et al., 2021; Ni et al., 2021), although one notable exception is Neelakantan et al. (2022). The recent work by Wang et al. (2023b) demonstrates that decoder-only LLMs can outperform frontier bidirectional embedding models (Wang et al., 2022; Ni et al., 2021; Chen et al., 2023) in retrieval and general-purpose embedding tasks.

---

*Correspondence to: Chankyu Lee <chankyul@nvidia.com>, Wei Ping <wping@nvidia.com>.

Table 1: Top MTEB leaderboard models as of ICLR submission date (2024-10-01). We use the original model names on the leaderboard for clarity.

| Embedding Task
Mertric | Retrieval (15)
nDCG@10 | Rerank (4)
MAP | Cluster. (11)
V-Meas. | PairClass. (3)
AP | Class. (12)
Acc. | STS (10)
Spear. | Summ.( 1)
Spear. | Avg. (56) |
|---|---|---|---|---|---|---|---|---|
| NV-Embed-v2 | **62.65** | 60.65 | **58.46** | 88.67 | **90.37** | 84.31 | 30.7 | **72.31** |
| Bge-en-icl (zero shot) | 61.67 | 59.66 | 57.51 | 86.93 | 88.62 | 83.74 | 30.75 | 71.24 |
| Stella-1.5B-v5 | 61.01 | 61.21 | 57.69 | 88.07 | 87.63 | 84.51 | 31.49 | 71.19 |
| SFR-Embedding-2R | 60.18 | 60.14 | 56.17 | 88.07 | 89.05 | 81.26 | 30.71 | 70.31 |
| Gte-Qwen2-7B-instruct | 60.25 | 61.42 | 56.92 | 85.79 | 86.58 | 83.04 | 31.35 | 70.24 |
| NV-Embed-v1 | **59.36** | 60.59 | 52.80 | 86.91 | 87.35 | 82.84 | 31.2 | **69.32** |
| Bge-multilingual-gemma2 | 59.24 | 59.72 | 54.65 | 85.84 | 88.08 | 83.88 | 31.2 | 69.88 |
| Voyage-large-2-instruct | 58.28 | 60.09 | 53.35 | 89.24 | 81.49 | 84.58 | 30.84 | 68.28 |
| SFR-Embedding | 59.00 | 60.64 | 51.67 | 88.54 | 78.33 | 85.05 | 31.16 | 67.56 |
| GritLM-7B | 57.41 | 60.49 | 50.61 | 87.16 | 79.46 | 83.35 | 30.37 | 66.76 |
| E5-mistral-7b-instruct | 56.9 | 60.21 | 50.26 | 88.34 | 78.47 | 84.66 | 31.4 | 66.63 |
| Text-embed-3-large (OpenAI) | 55.44 | 59.16 | 49.01 | 85.72 | 75.45 | 81.73 | 29.92 | 64.59 |

In this work, we introduce `NV-Embed`, a generalist embedding model that significantly enhances the performance of decoder-only LLMs for embedding and retrieval tasks. Specifically, we make the following contributions:

1. For model architecture, we propose a novel *latent attention layer* to obtain pooled embeddings for a sequence of tokens. In contrast to the popular average pooling in bidirectional embedding models (e.g., Wang et al., 2022) and last `<EOS>` token embedding in decoder-only LLMs (Neelakantan et al., 2022; Wang et al., 2023b), our proposed pooling technique consistently improves accuracy of retrieval and other downstream tasks. To further enhance representation learning, we remove causal attention mask during contrastive training of decoder-only LLM, resulting in solid improvements. Our design is simpler yet more effective compared to related work (BehnamGhader et al., 2024; Muennighoff et al., 2024), which involves an additional training phase with masked token prediction or a mixed training objective.

2. For model training, we introduce a two-stage contrastive instruction-tuning method, starting with the pretrained Mistral-7B (Jiang et al., 2023). In the first stage, we apply contrastive training with instructions on retrieval datasets, utilizing in-batch negative and curated hard-negative examples. In the second stage, we blend carefully curated non-retrieval datasets into the stage-one training data. Since in-batch negative samples are misleading for non-retrieval tasks in some cases, we disable in-batch negative training in stage two. This design not only improves the accuracy of classification, clustering, and semantic textual similarity tasks, but also surprisingly enhances retrieval performance. Note, our model is also not fine-tuned from existing embedding models[1].

3. Training data is one of the most crucial factors in achieving state-of-the-art results. We provide a detailed recipe on the curation of training datasets, including dataset-specific information, the positive-aware hard-negative mining technique to enhance contrastive training, the synthetic data generation and example-based multi-class labeling. This enables the community to easily reproduce and even surpass our model, ultimately advancing the development of the embedding models.

4. Our `NV-Embed`-v1 model obtained the No.1 position on the Massive Text Embedding Benchmark (MTEB) (as of May 24, 2024) (Muennighoff et al., 2022) across 56 embedding tasks. By improving the curation of the training data, `NV-Embed`-v2 model set a new record high score of **72.31** and reclaimed the No. 1 spot (as of Aug 30, 2024) on the highly competitive MTEB leaderboard, further demonstrating the sustained effectiveness of our approach. Note that our model also attains the highest scores in 15 retrieval tasks (commonly referred to as BEIR (Thakur et al., 2021)), 11 clustering tasks, and 12 classification tasks in the MTEB benchmark. See Table 1 for detailed information. Additionally, it secured the highest scores in Long Doc section and the second scores in QA section on the AIR-Benchmark which covers a range of out-of-domain information retrieval topics beyond those in MTEB.

5. We study the model compression techniques, including pruning, quantization and knowledge-distillation, for LLM-based embedding models. Through the comparison with smaller embedding models directly built on Llama3.2-3B, Qwen2.5-3B, and Minitron-4B, we demonstrate that our model compression approach achieves superior accuracy and quantization robustness.

We organize the rest of the paper in the following. In § 2, we discuss the related work. We present the architectural and training method in § 3. We provide detailed recipe of training data curation in § 4. We present the experiment results in § 5 and conclude the paper in § 6. Model compression techniques and results are presented in § A due to the page limit. AIR-bench results are shown in § B.

---

[1] For example, SFR-Embedding and Linq-Embed are fine-tuned from E5-mistral-7b-instruct.

## 2 RELATED WORK

### 2.1 BIDIRECTIONAL EMBEDDING MODELS

BERT (Devlin et al., 2018) or T5 (Raffel et al., 2020)-based embedding models have long been the dominant approaches for general-purpose embedding tasks. Early examples include Sentence-BERT (Reimers & Gurevych, 2019) and SimCSE (Gao et al., 2021), which finetune BERT on natural language inference (NLI) datasets. In general, these embedding models are first initialized from pre-trained BERT (Wang et al., 2022; Izacard et al., 2021) or T5 encoders (Ni et al., 2021). Then, they are further pre-trained with contrastive learning on curated unsupervised (Izacard et al., 2021) or weakly-supervised text pairs (Wang et al., 2022). Finally, the embedding models (Li et al., 2023; Wang et al., 2022; Ni et al., 2021; Chen et al., 2023) are fine-tuned on a variety of supervised data, including MS MARCO (Nguyen et al., 2016), for retrieval and other downstream tasks. Note that all the state-of-the-art embedding models are trained in this supervised manner. Some of the most recent frontier models in this category include mxbai-embed-large-v1 (Lee et al., 2024b) (MTEB: 64.68), UAE-Large-V1 (Li & Li, 2023) (MTEB: 64.64), and voyage-large-2-instruct (Voyage-AI, 2024) (MTEB: 68.28).

### 2.2 DECODER-ONLY LLM-BASED EMBEDDING MODELS

Decoder-only LLMs (Brown et al., 2020) were believed to underperform bidirectional models on general-purpose embedding tasks for years, because: *i*) unidirectional attention limits the representation learning capability, and *ii*) the scaling of LLMs leads to very high-dimension embeddings, which may suffer from the *curse of dimensionality*.

The early work by Neelakantan et al. (2022) initializes embedding models using pre-trained, decoder-only GPT-3 models (Brown et al., 2020) and applies continued contrastive training. The hidden state from the final layer, corresponding to the special token *<EOS>* at the end of the sequence, is used as the embedding for the input sequence. Its latest successor, text-embedding-3-large, achieves an MTEB score of 64.59 (OpenAI, 2024). Most recently, E5-Mistral (Wang et al., 2023b) (MTEB: 66.63) applies contrastive learning with task-specific instructions on Mistral 7B (Jiang et al., 2023). It begins to outperform the state-of-the-art bidirectional models on comprehensive embedding benchmarks (Muennighoff et al., 2022) by utilizing a massive amount of synthetic data from the proprietary GPT-4 model. LLM2Vec (BehnamGhader et al., 2024) (MTEB score: 65.01) tries to build the embedding model from LLMs while only using public available data, but it is still worse than E5-Mistral.

Given the success of E5-Mistral, SFR-Embedding-Mistral (Meng et al., 2024b) (MTEB: 67.56) and SFR-Embedding-2R (Meng et al., 2024a) (MTEB: 70.31) further fine-tunes this model on the blend of non-retrieval and retrieval datasets for improved accuracy on both tasks, which is closely related to our NV-Embed. However, there are the following key differences: 1) NV-Embed is trained from scratch on Mistral 7B LLM directly using public available data, and not dependent on other embedding model or proprietary synthetic data. Consequently, we introduce a new architecture that eliminates unnecessary causal attention mask and further improves the sequence pooling mechanism with latent attention layer. 2) SFR-Embedding-Mistral uses task-homogeneous batching, which constructs batches consisting exclusively of samples from a single task. In contrast, our NV-Embed uses well-blended batches consisting samples from all tasks to avoid potential "zigzag" gradient updates, which leads to a new record high score on both full MTEB and retrieval tasks compared to SFR-Embedding-Mistral.

Over the past year, MTEB has become one of the most competitive leaderboards across all AI categories, leading to significantly increased competition among participants. Many of the recent top-performing models (e.g., stella-1.5B-v5, gte-Qwen2-7B-instruct, bge-multilingual-gemma2, voyage-large-2-instruct, and text-embed-3-large) have not disclosed key technical details necessary for reproduction, particularly the blend of training data used. Among the recently disclosed works, GritLM (Muennighoff et al., 2024) (MTEB: 65.66) unifies text embedding and generation into a single LLM model. In addition, bge-en-icl (Li et al., 2024) (MTEB: 71.24) enhances query embeddings by introducing few-shot examples on the query side, utilizing the in-context learning (ICL) capabilities in text embedding tasks. This approach introduces an overhead by supplying task-relevant examples to the query during the training process. To maintain zero-shot evaluation accuracy, both zero-shot

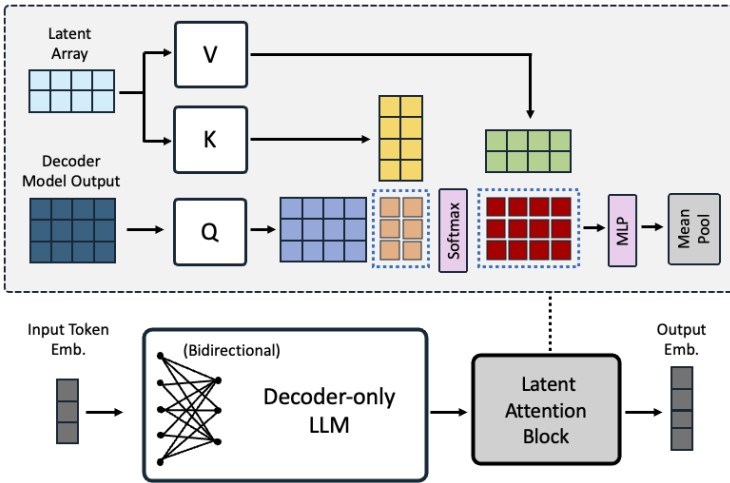

Figure 1: Proposed architecture design comprising of decoder-only LLM followed by latent attention layer. Latent attention layer functions as a form of cross-attention where the decoder-only LLM output serves as queries ($Q$) and trainable latent array passes through the key-value inputs, followed by MLP. Blue dotted lines indicate the two matrix multiplications involved in QKV-attentions.

and few-shot samples are included during training. In our paper, we focus on comparing the zero-shot evaluation accuracy of the bge-en-icl model to ensure the fair comparisons during the evaluation phase.

Another area of research focuses on improving data curation processes to enhance the accuracy of fine-tuning retrieval embedding models. Gecko (Lee et al., 2024a) (MTEB: 66.31) attempts to distill a smaller bidirectional embedding model from a decoder-only LLM (Gemini et al., 2023) by generating synthetic paired data. It refines the data quality by retrieving a set of candidate passages for each query and relabeling the positive and hard negative passages using the LLM. Linq-embed-mistral (Kim et al., 2024) utilized LLMs to refine data by generating, filtering, and mining negative samples. Meanwhile, NV-Retriever (Moreira et al., 2024) introduced a positive-aware hard-negative mining technique that considers positive relevance scores to more effectively eliminate false negatives. In this work, we apply this positive-aware hard-negative technique to curate the samples and enhance the contrastive training.

## 3 METHODS

In this section, we describe our architecture designs and two-stage instruction-tuning method.

### 3.1 BIDIRECTIONAL ATTENTION

The causal attention mask in decoder-only LLMs is introduced for next-token prediction task (Vaswani et al., 2017). In principle, causal mask in decoder blocks prevents information leakage by allowing the decoder to attend only to previous positions during auto-regressive text generation. However, it is observed that unidirectional attention limits the model's representation power, as evidenced by the poor performance of GPT models compared to similarly sized BERT or T5 models on natural language understanding benchmarks (e.g., Wang et al., 2019). In recent, LLM2Vec (BehnamGhader et al., 2024) introduces additional training phase with a specially designed masked token prediction to warm-up the bidirectional attention. GRIT (Muennighoff et al., 2024) utilizes a hybrid objective with both bidirectional representation learning and causal generative training. In contrast, we simply remove the causal attention mask of decoder-only LLM during the contrastive learning and find it works compellingly well as demonstrated by our results. As a result, we go with simple solution.

## 3.2 LATENT ATTENTION LAYER

There are two popular methods to obtain the embedding for a sequence of tokens: *i)* mean pooling, and *ii)* the last `<EOS>` token embedding. Previous bidirectional embedding models typically use mean pooling (Wang et al., 2022; Izacard et al., 2021), while the last `<EOS>` token embedding is more popular for decoder-only LLM based embedding models. However, both methods have certain limitations. Mean pooling simply takes the average of token embeddings and may dilute the important information from key phrases, meanwhile the last `<EOS>` token embedding may suffer from *recency bias*, relying heavily on the output embedding of last token.

In this work, we propose a latent attention layer inspired by Jaegle et al. (2021) to achieve more expressive pooling of the sequences for general-purpose embedding tasks. Specifically, we denote the last layer hidden from decoder as the query $Q \in \mathbb{R}^{l \times d}$, where $l$ is the length of sequence, and $d$ is the hidden dimension. They are sent to attend the latent array $K = V \in \mathbb{R}^{r \times d}$, which are *trainable* "dictionary" used to obtain better representation, where $r$ is the number of latents in the dictionary. The output of this cross-attention is $O \in \mathbb{R}^{l \times d}$,

$$O = \text{softmax}(QK^T)V \tag{1}$$

which is followed by a regular MLP consists of two linear transformations with a GELU activation in between. Our model uses latent attention layer with $r$ of 512 and the number of heads as 8 for multi-head attention. Finally, we apply mean pooling after MLP layers to obtain the embedding of whole sequences. See Figure 1 for an illustration. It is worth mentioning here that our approach follows the spirit of dictionary learning to obtain better representation (e.g., Wang et al., 2018), which is different from the Perceiver IO architecture. We compare the proposed *latent attention layer* with normal self-attention and find consistent improvements in our ablation study.

## 3.3 TWO-STAGE INSTRUCTION-TUNING

Instruction-tuning has been widely applied for training LLM to follow instructions (Wei et al., 2021; Ouyang et al., 2022) and to perform retrieval-augmented generation (Wang et al., 2023a; Liu et al., 2024). It has also been recently applied for training retrievers and general-purpose embedding models that can adapt their output embeddings with different instructions and task types (Asai et al., 2022; Wang et al., 2023b).

To obtain a generalist embedding model that can appropriately perform on retrieval and non-retrieval tasks (e.g., classification, clustering), we need take the characteristics of different tasks into account. For example, the use of in-batch negatives has been demonstrated to be highly efficient for training dense-embedding-based retrievers (e.g., Karpukhin et al., 2020), because it allows to reuse the computation and effectively train on $B^2$ question/passage pairs for each mini-batch with only $B$ questions and corresponding positive passages. However, applying in-batch negatives trick can mislead the embedding model for classification or clustering task, as the "passages" in the mini-batch may come from the the class and are not negatives.

Given these considerations, we introduce a two-stage instruction tuning method which first conducts contrastive training with instructions on a variety of retrieval datasets (details are in section 4.1), utilizing in-batch negatives and curated hard-negative examples. In the second stage, we perform contrastive instruction-tuning on a combination of retrieval and non-retrieval datasets (details are in section 4.2) without applying the trick of in-batch negatives. It is worth mentioning here that retrieval task presents greater difficulty compared to the other tasks so that our training strategy focuses on fine-tuning the model for retrieval initially. In second stage, we blend the remaining embedding tasks into the instruction-tuning.

## 4 TRAINING DATA

For training data, we employ public retrieval and non-retrieval datasets and synthetically generated samples to demonstrate our model's capability in embedding tasks. Our training procedure incorporates both retrieval and non-retrieval tasks including classification, clustering, and semantic textual similarity datasets.

Given a relevant query-document pair, the instructed query follows the instruction template as follows:

$$q_{\text{inst}}^+ = \texttt{Instruct:\{task\_definition\} Query:} q^+ \qquad (2)$$

The instruction templates for each {task_definition} are provided in Table 12 for training and Table 13 for evaluation. Note, we mask out the instruction tokens in the output embeddings during both training and evaluation, although they still impact the output due to self-attention. We do not add any instruction prefix to document corpus.

## 4.1 PUBLIC RETRIEVAL DATASETS

We adopt the retrieval datasets as follows: MSMARCO (Bajaj et al., 2016), HotpotQA (Yang et al., 2018), Natural Question (Kwiatkowski et al., 2019), PAQ (Lewis et al., 2021), Stack Exchange (Stack-Exchange-Community, 2023), Natural Language Inference (Group et al., 2022), SQuAD (Rajpurkar et al., 2016), ArguAna (Wachsmuth et al., 2018), BioASQ (Tsatsaronis et al., 2015), FiQA (Maia et al., 2018), FEVER (Thorne et al., 2018), HoVer (Jiang et al., 2020), SciFact (Wadden et al., 2022), NFCorpus, MIRACL (Zhang et al., 2023) and Mr.TyDi (Zhang et al., 2021).

It is important to note that certain datasets (e.g., MSMARCO) are training splits of the MTEB Benchmark, which we follow the existing practices established by leading generalist embedding models (Meng et al., 2024b; Wang et al., 2023b; BehnamGhader et al., 2024; Muennighoff et al., 2024). Table 12 further provides the number of samples used for training. We demonstrate the zero-shot generalization capability of NV-Embed on AIR-bench in B.

### 4.1.1 HARDNEGATIVE MINING TECHNIQUE

Embedding models are trained using contrastive learning (Gao et al., 2021), aiming to increase the similarity between the embeddings of a query and its relevant passages (positives) while reducing the similarity with irrelevant passages (negatives). Public retrieval datasets typically only contains the positive query-passage pairs but do not contain its own hardnegatives, making it necessary to mine of such negative examples. To address this, we apply the recently proposed positive-aware hard-negative technique (Moreira et al., 2024) that considers the positive relevance scores for better false negatives removal. Following the ablation studies in Moreira et al. (2024), we use E5-mistral-7b-instruct (Wang et al., 2023b) as a teacher retrieval model to identify the optimal hardnegative passages relevant to the query. We set the maximum threshold for negative scores based on a percentage of the positive score (TopKPercPos) with a 95% margin, described as follows: max_negative_score_threshold = pos_score * percentage_margin.

## 4.2 PUBLIC NON-RETRIEVAL DATASETS

Besides retrieval datasets, we utilize public non-retrieval datasets mainly from three sub-tasks in MTEB benchmark: classification, clustering and semantic similarity (STS). We pre-process the format of these datasets to become the compatible with retrieval datasets for contrastive training: query $q^+$, positive document $d^+$ and hard negative documents $\{d_0^-, ..., d_n^-\}$.

For classification, we utilize the English training splits of various datasets from MTEB Huggingface datasets (Muennighoff et al., 2022; Lhoest et al., 2021). The classification datasets that we use are as follows: AmazonReviews (McAuley & Leskovec, 2013a), AmazonCounterfactual (O'Neill et al., 2021), Banking77 (Casanueva et al., 2020), Emotion (Saravia et al., 2018), IMDB (Maas et al., 2011), MTOPDomain/MTOPIntent (Li et al., 2021), ToxicConversations (Adams et al., 2019), TweetSentimentExtraction (Maggie, 2020), AmazonPolarity (McAuley & Leskovec, 2013b), MassiveScenario/MassiveIntent (FitzGerald et al., 2022). For the Emotion and AmazonCounterfactual classification datasets we use BM25 (Robertson et al., 2009) similarity thresholds to filter out training data that is similar to the MTEB evaluation set.

For clustering datasets, we utilize the raw_arxiv, raw_biorxiv and raw_medrxiv datasets from MTEB Huggingface datasets, TwentyNewsgroups (Lang, 1995), Reddit (Geigle et al., 2021), StackExchange (Geigle et al., 2021), RedditP2P (Reimers, 2021b) and StackExchangeP2P (Reimers, 2021a) We filter out any training data that match the MTEB evaluation set.

The classification and clustering datasets provide examples and corresponding class/cluster labels. The example texts extracted from the appropriate *text/title/abstract* field are used for the query

$q^+$. For binary classification tasks the label texts are used as documents $d^+, d^-$. For multi-class classification and clustering tasks, a randomly sampled example from the ground-truth class/cluster is used for the positive document $d^+$ and randomly sampled examples from other classes/clusters are used for negative documents $d_k^-$. We will present ablation experiments supporting this approach in section 5.2.4.

For semantic textual similarity datasets, we use the training splits of three semantic similarity datasets STS12 (Agirre et al., 2012), STS22 (Chen et al., 2022), STS-Benchmark (Cer et al., 2017) from MTEB Huggingface datasets. For any pair of texts with associated relevance scores $(t_a, t_b, score)$, we create two examples $(q^+ = t_a, d^+ = t_b)$ and $(q^+ = t_b, d^+ = t_a)$ if $score \geq 4$. We mine the hard negatives $d_k^-$ from the pool of other texts using the same technique as section 4.1.1. Task instructions are appended to $d^+, d^-$ since they are symmmetric with the query.

### 4.3 SYNTHETIC TASKS DATASET

Due to the limited variety of subjects and tasks in public training datasets, the available instruction templates for training are also restricted. To enhance task-wise generalization, we employ the Mixtral-8x22B-Instruct-v0.1 model (MistralAI) to create a dataset consisting of 120,000 synthetic examples across 60,000 synthetic tasks. Following a two-step prompting approach proposed by E5-mistral-7b-instruct (Wang et al., 2023b), we adjust the prompts for Mixtral-8x22B-Instruct-v0.1 and English text. We generate only the short-long, long-short, and short-short examples (40,000 of each), as we use public STS datasets and do not assess bitext retrieval tasks. Example prompts for synthetic data generation can be found in Appendix 15 and 16.

## 5 EXPERIMENTS

Training and inference experiment details are illustrated in Appendix C.

### 5.1 MTEB RESULTS

We evaluate the proposed `NV-Embed` model on the full MTEB benchmark (Muennighoff et al., 2022) across 56 tasks. Table 1 summarizes averaged MTEB scores for seven sub-category tasks compared to frontier models on MTEB leaderboard[2]. Our initial model, namely `NV-Embed-v1` get the score of 69.32 and obtain the No.1 position on the MTEB as of May 24, 2024 (detailed benchmark scores available in Table 2). We then further improve the model through the curation of training dataset, including adding more retrieval datasets, applying positive-aware hard-negative mining technique, using synthetic data generation process and constructing example-based multi-class labels. As a result, our `NV-Embed-v2` model sets a new record high score of **72.31** and reclaimed No.1 (as of Aug 30, 2024) on highly competitive MTEB leaderboard, further highlighting the sustained effectiveness of the proposed methods. In following sub-section 5.2, we will present ablation studies on design choices regarding the model architecture, training algorithm and the curation of training data.

Based on quantitative leaderboard results, we compare our `NV-Embed` with the recent frontier embedding models. The e5-mistral-7b-instruct (Wang et al., 2023b) and google-gecko (Lee et al., 2024a) utilize proprietary synthetic data to train their model in a single stage manner. In contrast, we recognize that retrieval task presents greater difficulty compared to the other embedding tasks and prioritizes our training strategy on fine-tuning the model for retrieval first, followed by blending the remaining sub-tasks into instruction-tuning, leading to substantially improved BEIR and overall MTEB results.

SFR-Embedding-2R (Meng et al., 2024b) demonstrates competitive scores on the MTEB (70.31) and BEIR (60.18) benchmarks by continuing to finetune the e5-mistral-7b-instruct model (Wang et al., 2023b). However, it remains largely constrained by the architectural limitations of its parent model, such as the causal attention mask and the last token pooling method. In contrast, our `NV-Embed` model is trained starting from the Mistral 7B LLM (Jiang et al., 2023) rather than finetuning e5-mistral-7b-instruct (Wang et al., 2023b). It features a new architecture that removes the unnecessary causal attention mask and further improves the sequence pooling mechanism with a latent attention layer. Table 3 and 14 provides a detailed scores of BEIR and MTEB benchmarks.

---

[2]https://github.com/embeddings-benchmark/mteb

Table 2: Averaged MTEB scores on seven tasks after first and second stage training using only the publically available data and before applying the positive-aware hardnegative mining, synthetic data and example-based multi-class labeling. The averaged score **69.32** corresponds to NV-Embed-v1.

| | First stage training | | | | | | | |
|---|---|---|---|---|---|---|---|---|
| Pool Type | EOS | | Mean | | Latent-attention | | Self-attention | |
| Mask Type | bidirect | causal | bidirect | causal | bidirect | causal | bidirect | causal |
| Retrieval(15) | 57.70 | 56.42 | 58.42 | 57.55 | **59.00** | 57.65 | 57.89 | 57.21 |
| Rerank (4) | 59.76 | 57.21 | 60.02 | 59.35 | 59.59 | 59.72 | 59.73 | 59.51 |
| Clustering (11) | 44.75 | 40.83 | 45.97 | 45.42 | 45.44 | 45.61 | 45.19 | 45.07 |
| PairClass. (3) | 86.17 | 83.63 | 87.45 | 84.46 | 87.59 | 82.02 | 86.51 | 85.74 |
| Classification (12) | 73.17 | 69.22 | 74.62 | 72.48 | 73.93 | 72.74 | 73.54 | 73.32 |
| STS (10) | 74.96 | 73.45 | 77.47 | 73.60 | 79.07 | 78.65 | 76.89 | 77.55 |
| Summar. (1) | 29.28 | 28.4 | 29.72 | 30.89 | 30.16 | 30.94 | 30.22 | 31.59 |
| **Average (56)** | 62.68 | 60.06 | 64.00 | 62.32 | 64.18 | 63.39 | 63.27 | 63.11 |

| | Second stage training | | | | | | | |
|---|---|---|---|---|---|---|---|---|
| Pool Type | EOS | | Mean | | Latent-attention | | Self-attention | |
| Mask Type | bidirect | causal | bidirect | causal | bidirect | causal | bidirect | causal |
| Retrieval (15) | 58.39 | 56.59 | 58.71 | 57.88 | **59.36** | 58.33 | 58.64 | 57.71 |
| Rerank (4) | 60.37 | 59.23 | 60.77 | 60.27 | 60.54 | 60.57 | 60.5 | 60.38 |
| Clustering (11) | 51.43 | 49.81 | 52.80 | 51.58 | 52.80 | 51.7 | 53.34 | 51.51 |
| PairClass. (3) | 84.06 | 80.99 | 87.45 | 82.89 | 86.91 | 83.45 | 86.12 | 84.44 |
| Classification (12) | 85.85 | 85.04 | 87.06 | 86.08 | 87.35 | 86.58 | 86.76 | 86.25 |
| STS (10) | 79.55 | 79.12 | 82.53 | 81.74 | 82.84 | 81.94 | 82.38 | 81.52 |
| Summar. (1) | 30.36 | 29.12 | 30.49 | 31.82 | 31.20 | 31.87 | 30.105 | 31.4 |
| **Average (56)** | 67.85 | 66.50 | 68.97 | 68.13 | **69.32** | 68.47 | 69.10 | 68.16 |

Table 3: Averaged MTEB scores on seven embedding tasks after two stage training after applying the positive-aware hardnegative mining, synthetic data and example-based multi-class labeling. Note, the averaged score **72.31** corresponds to NV-Embed-v2.

| Pool Type | EOS | | Mean | | Latent-attention | | Self-attention | |
|---|---|---|---|---|---|---|---|---|
| Mask Type | bidirect | causal | bidirect | causal | bidirect | causal | bidirect | causal |
| Retrieval (15) | 62.13 | 60.30 | 61.81 | 61.01 | **62.65** | 61.15 | 61.17 | 60.53 |
| Rerank (4) | 60.02 | 59.13 | 60.65 | 59.10 | 60.65 | 59.36 | 60.67 | 59.67 |
| Clustering (11) | 58.24 | 57.11 | 57.44 | 57.34 | **58.46** | 57.80 | 58.24 | 57.11 |
| PairClass. (3) | 87.69 | 85.05 | 87.35 | 87.35 | 88.67 | 87.22 | 87.69 | 85.05 |
| Classification (12) | 90.10 | 90.01 | 89.49 | 89.85 | **90.37** | 90.49 | 90.10 | 90.01 |
| STS (10) | 82.27 | 81.65 | 84.35 | 84.35 | 84.31 | 84.13 | 84.22 | 83.81 |
| Summar. (1) | 30.25 | 32.75 | 30.75 | 30.88 | 30.70 | 30.90 | 30.93 | 31.36 |
| **Average (56)** | 71.63 | 70.85 | 71.71 | 71.38 | **72.31** | 71.61 | 71.61 | 70.6 |

## 5.2 ABLATION STUDY

We conduct ablation studies to compare several training, architecture and data curation design choices: two-stage training, bidirectional attention, latent-attention pooling method, synthetic data and example-based multi-class labeling.

### 5.2.1 TWO-STAGE TRAINING

We compare the two-stage and single-stage training with and without the use of the in-batch negative technique, as shown in Table 4. We observe that our proposed two-stage training surpasses single-stage training because it allows the use of beneficial in-batch negatives for retrieval tasks in the first stage, while disabling the in-batch technique for non-retrieval tasks in the second stage. In contrast, single-stage training with in-batch negatives leads to significantly lower MTEB performance, especially in the classification sub-task. This accuracy degradation occurs because many classification tasks involve few-class labels (such as binary labels like True/False), meaning that the inbatch negative labels in the batch can actually be the positive label. While single-stage training without in-batch negatives produces more comparable results (MTEB scores: 72.31 for two-stage training vs. 71.94 for single-stage without in-batch), two-stage training significantly outperforms in the retrieval sub-tasks (BEIR scores: 62.65 for two-stage training vs. 61.37 for single-stage without in-batch). It is worth

Table 4: Averaged MTEB scores on ablation studies for `NV-Embed-v2`: two stage training, multi-class data labeling, positive-aware hardnegative mining and synthetically generated dataset. In the third part of the table, HN represents hardnegative mining technique, AD means adding public retrieval datasets and SD refers to adding synthetically generated data. In the fourth part of the table, we also include `NV-Embed-v1`, which omits HN, AD, and SD in stage-one training and uses a label-based approach in stage-two training.

| Section 5.3.1 Two stage training | | | | | | | | |
|---|---|---|---|---|---|---|---|---|
| Embedding Task | Retrieval | Rerank | Cluster. | PairClass. | Class. | STS | Summ. | Avg. |
| Single Stage (Inbatch Enabled) | 61.25 | 60.64 | 57.67 | 87.82 | 86.6 | 83.7 | 30.75 | 70.83 |
| Single Stage (Inbatch Disabled) | 61.37 | 60.81 | 58.31 | 88.3 | 90.2 | 84.5 | 30.96 | 71.94 |
| **Two Stage Training** | **62.65** | 60.65 | 58.46 | 88.67 | 90.37 | 84.31 | 30.70 | **72.31** |
| **Reversed Two Stage** | **61.91** | 60.98 | 58.22 | 88.59 | 90.26 | 83.07 | 31.28 | **71.85** |

| Section 5.3.4 Multi-class Classification and Clustering Labels in stage-two training | | | | | | | | |
|---|---|---|---|---|---|---|---|---|
| Embedding Task | Retrieval | Rerank | Cluster. | PairClass. | Class. | STS | Summ. | Avg. |
| Label-based approach | 62.40 | 59.7 | 53.04 | 88.04 | 89.17 | 84.25 | 30.77 | 70.82 |
| **Example-based approach** | 62.65 | 60.65 | **58.46** | 88.67 | **90.37** | 84.31 | 30.70 | **72.31** |

| Section 5.3.5 Hard-negative mining and Synthetically Generated Dataset in stage-one training | | | | | | | | |
|---|---|---|---|---|---|---|---|---|
| Embedding Task | Retrieval | Rerank | Cluster. | PairClass. | Class. | STS | Summ. | Avg. |
| [S0] Without HN, Without AD, Without SD | 59.22 | 59.85 | 57.95 | 85.79 | 90.71 | 81.98 | 29.87 | 70.73 |
| [S1] With HN, Without AD, Without SD | 61.52 | 59.80 | 58.01 | 88.56 | 90.31 | 84.26 | 30.36 | 71.83 |
| [S2] With HN, With AD, Without SD | 62.28 | 60.45 | 58.16 | 88.38 | 90.34 | 84.11 | 29.95 | 72.07 |
| **[S3] With HN, With AD, With SD** | 62.65 | 60.65 | 58.46 | 88.67 | 90.37 | 84.31 | 30.70 | **72.31** |

| NV-Embed-v1 | | | | | | | | |
|---|---|---|---|---|---|---|---|---|
| Label-based approach + [S0] | 59.36 | 60.59 | 52.80 | 86.91 | 87.35 | 82.84 | 31.2 | 69.32 |

highlighting here that the retrieval is considered the most crucial sub-category for the advancement of RAG technology across the MTEB embedding tasks.

Lastly, we explore another research question: what happens if the order of two-stage training is reversed? To examine this, we further finetune the Single Stage (Inbatch disabled) model using only the retrieval datasets with enabling the inbatch negative technique and present the MTEB results in Table 4. While the retrieval score increased from 61.37 to 61.91 after the reversed two-staged training, it remains lower than the retrieval score of 62.65 achieved with our proposed two-stage training method. Furthermore, the scores on other embedding tasks, such as Clustering and STS, declined compared to the Single Stage (Inbatch disabled) approach. Consequently, the overall MTEB score for Reversed Two Stage (score: 71.85) is lower than our proposed Two-Stage Training (score: 72.31) as well as the Single Stage with Inbatch disabled (score: 71.94).

### 5.2.2 CAUSAL ATTENTION VS. BIDIRECTIONAL ATTENTION

To examine the impact of self-attention masks in decoder-only LLM models for embedding applications, we conducted experiments comparing bidirectional and causal mask types. As illustrated in Tables 2 and 3, the bidirectional mask consistently outperforms the causal mask based on the average MTEB scores across 56 tasks for all pooling types. This indicates that embeddings generated with causal attention masks are significantly less effective than those produced with bidirectional attention masks.

### 5.2.3 POOLING METHODS

To examine the impact of different pooling methods on embedding models, we conducted experiments comparing `<EOS>`-last, mean, latent-attention, and self-attention pooling types. As depicted in Tables 2 and 3, mean pooling consistently outperforms `<EOS>`-last token embedding based on the average MTEB scores across 56 tasks. This difference may be due to the last `<EOS>` token embedding being influenced by *recency bias*, showing an excessive dependence on the output of the final token.

To enhance performance beyond mean pooling, we experimented with adding the proposed latent-attention or self-attention layer (both followed by MLP) before mean pooling to address the issue of important information from key phrases being diluted. According to Tables 2, self-attention does not provide additional accuracy improvements for the embedding capabilities of decoder-only LLMs (i.e., mean pooling 68.97 vs. self-attention 69.10 on MTEB tasks). It even slightly reduces accuracy

on 15 retrieval tasks (i.e., mean pooling 58.71 vs. self-attention 58.64). Table 3 also shows the similar trends of `NV-Embed`-v2. This is not surprising, as the LLM already has many self-attention layers to learn the representation, and adding an additional one does not bring significant additive value.

In contrast, the latent-attention layer proved beneficial for majority of embedding tasks, as shown in Table 2 and 3. Specifically, the nDCG@10 accuracy of the more challenging 15 retrieval tasks improved (i.e., mean pooling 61.82 vs. latent-attention 62.65) in Table 3. We hypothesize that this is due to the "dictionary learning" provided by the latent array, which offers more expressive representation. The latent-attention layer effectively learns output embedding representations from decoder-only LLMs, mitigating the information dilution caused by averaging the output embeddings.

### 5.2.4 MULTI-CLASS CLASSIFICATION AND CLUSTERING LABELS

We compare the effect of using two possible techniques for constructing positive and negative documents for multi-class classification and clustering tasks. In label-based approach, the ground-truth class/cluster label corresponding to the example in the query is used as the positive document, and other class/cluster labels are sampled for negative documents. In example-based approach, another example from the same class/cluster as the example in the query is used as the positive document, and examples from other clusters are sampled for negative documents. We use random sampling to get a broad coverage across labels and examples. In this work, all 11 clustering datasets and 5 muti-class classification datasets are constructed as example-based approach. As shown in Table 4, the example-based approach leads to significant improvements over the label-based approach for both classification and clustering. Table 5 further shows the detailed ablation study of label-based and example-based labels for classification and clustering multi-class samples.

Table 5: Ablation study on using class/cluster labels vs. sampled class/cluster examples as positive and negative documents for multi-class classification and clustering tasks.

| +/- Document Format | Labels | Examples |
|---|---|---|
| Emotion-Classification | 90.83 | 93.38 |
| MassiveIntent-Classification | 84.94 | 86.10 |
| MassiveScenario-Classification | 90.18 | 92.17 |
| MTOPDomain-Classification | 98.84 | 99.25 |
| MTOPIntent-Classification | 88.55 | 94.37 |
| Arxiv-Clustering-P2P | 53.01 | 55.80 |
| Arxiv-Clustering-S2S | 49.19 | 51.26 |
| Biorxiv-Clustering-P2P | 45.38 | 54.09 |
| Biorxiv-Clustering-S2S | 42.67 | 49.60 |
| Medrxiv-Clustering-P2P | 37.58 | 46.09 |
| Medrxiv-Clustering-S2S | 36.82 | 44.86 |
| Reddit-Clustering | 59.83 | 71.10 |
| Reddit-Clustering-P2P | 72.58 | 74.94 |
| StackExchange-Clustering | 79.37 | 82.10 |
| StackExchange-Clustering-P2P | 48.59 | 48.36 |
| TwentyNewsgroups-Clustering | 58.41 | 64.82 |
| **Average (16)** | **64.80** | **69.27** |

### 5.2.5 HARDNEGATIVE MINING AND SYNTHETICALLY GENERATED DATASET

We provide a step-by-step curation of training dataset, incorporating the hard negative mining technique (`S1`), additional public retrieval data (`S2`), and synthetically generated data (`S3`). As shown in Table 4, the first step of adding the hard negative mining technique significantly boosted retrieval accuracy, with the BEIR score increasing from 59.22 to 61.52. In the next step (`S2`), we included more public retrieval datasets (HoVer, SciFact, Nfcorpus, MIRACL, Mr.Tydi) followed by synthetically generated data. Adding the public retrieval datasets further increased the retrieval score by 0.7 points. Finally, incorporating the synthetic dataset (`S3`) leads to a modest improvement in the overall MTEB scores, raising them by 0.24 points.

## 6 CONCLUSION

We introduced the `NV-Embed` model, a decoder-only LLM designed to outperform existing bidirectional models in general-purpose text embedding tasks. For model architecture, we propose a latent attention layer to obtain expressive pooled embeddings and remove the unnecessary causal attention mask of decoder-only LLMs. For training algorithm, we introduce a two-stage contrastive instruction-tuning scheme to sequentially improve the embedding tasks. By leveraging carefully curated datasets, hard-negative mining, synthetic data generation and example-based multi-class labeling, our approach achieve the superior accuracy across diverse embedding tasks. As a result, the series of `NV-Embed` models achieved and maintained the No.1 ranking on the MTEB leaderboard and also demonstrated superior accuracy in out-of-domain tasks in AIR Benchmark.

## 7 ACKNOWLEDGMENT

We would like to extend our sincere gratitude to the NVIDIA Merlin team for their valuable collaboration and insightful discussions on building embedding and retriever models. We especially wish to thank Benedikt Schifferer, Gabriel de Souza P. Moreira, Radek Osmulski, Mengyao Xu, Ronay Ak, and Even Oldridge for providing the data from NV-Retriever (Moreira et al., 2024).

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

## A    COMPREHENSIVE STUDY OF MODEL COMPRESSION TECHNIQUES FOR NV-EMBED

Increasing computational and memory demands of LLM-based embedding model present the challenges for the deployment, limiting their scalability and accessibility. In this appendix section, we provide the analysis of post-training model compression techniques (i.e., pruning and quantization) for generalist embedding models. Our analysis demonstrates that these compression methods enhance the accuracy and robustness of LLM-based embedding models, surpassing the performance of smaller-sized embedding models based on Llama3.2-3B, Qwen2.5-3B and Minitron-4B.

In model compression process, we first perform pruning the NV-Embed-v2 model, reducing its size from 8 billion parameters to 3.5 billion (i.e., pruning the main decoder-only blocks and removing the latent attention block). Next, we apply quantization to lower its precision to 8-bit weights including integer and floating (E4M3, E5M2) formats. Finally, we perform continual re-training using fine-tuning (PEFT) method known as low-rank adaptation (LoRA) to restore the model's accuracy. For evaluation, we evaluate our model on MTEB benchmark (Muennighoff et al., 2022).

### A.1    PRUNING

In order to find better pruning techniques, we apply three methods (magnitude-based, WANDA(Sun et al., 2023), SparseGPT(Frantar & Alistarh, 2023)) for semi-structured (2:4 and 4:8) and unstructured approaches. Note, unstructured pruning strategy removes the network elements from individual weights, while the structured strategy removes the blocks of nonzero weights in higher granularity ways such as row/columns of weight metrics. Semi-structured is the hardware friendly way (N:M sparsity), ensuring that N weights remain non-zero within every group of M weights. For example, 4:8 semi-structured pruning prunes four out of every eight elements in a weight tensor. This semi-structured sparsity reduces the size of the weight matrices and computational cost, while maintaining certain regularity for efficient hardware utilization. The literature presents various criteria for determining which weights to prune. The simplest approach is magnitude-based pruning, which retains weights with higher absolute values and removes the rest. Another approach is WANDA (Sun et al., 2023) which introduces a pruning technique that considers both weights and activations. SparseGPT (Frantar & Alistarh, 2023) identifies the non-critical connections by utilizing the approximate hessian based optimization method.

Table 6 summarizes the averaged MTEB scores for different model pruning, respectively. Among these techniques, SparseGPT generally delivers the best results, while magnitude-based and WANDA methods produce comparable performance both during pruning and after retraining as shown in Table 6. Notably, semi-structured (2:4) pruning yields the lowest scores but demonstrates the greatest accuracy recovery following retraining for MTEB benchmarks. Based on these findings, we focus on SparseGPT pruning for subsequent ablation studies.

Table 6: Pruning - MTEB benchmark

| Pruning Criterion | | Semi-structured | | Unstructured |
|---|---|---|---|---|
| | | 2:4 | 4:8 | |
| Magnitude | Pruning | 64.62 | 67.6 | 69.18 |
| | Re-train | 69.96 | 70.46 | 70.84 |
| Wanda | Pruning | 64.26 | 67.87 | 70.19 |
| | Re-train | 69.74 | 70.42 | 70.81 |
| SparseGPT | Pruning | 68.48 | 70.11 | 71.33 |
| | Re-train | 70.41 | 70.9 | 71.18 |

### A.2    KNOWLEDGE DISTILLATION

In traditional accuracy recovery approaches after model compression, ground truth labels are utilized for continual retraining. To improve this retraining process, we leverage a knowledge distillation loss term, where the uncompressed model serves as the teacher, transfering the knowledge of the more advanced teacher model to a smaller and simpler student model. To enable the student model mimic the teacher's behavior, we introduce mean-squared error losses for both the output state ($S_o$) and the intermediate states ($S_1 - S_{o-1}$).

For this knowledge distillation process, we use the the uncompressed embedding model serves as the teacher, while the compressed version acts as the student. We remove the latent attention block and compensate the accuracy degradation with knowledge distillation. The knowledge distillation loss is defined as $L_{kd} = \sum_{n=1}^{O-2}[MSE(S_s^n, S_t^n)] + MSE(S_s^{O-1}, S_t^O)$ where $L_{kd}$ is knowledge distillation loss, O is the number of layers, n is layer number, MSE represents the mean-squared function, $S_s$ is student state and $S_t$ is the teacher state. Based on this, the total loss function is sum of contrastive and knowledge distillation loss as: $L_{total} = L_{contrastive} + \alpha \times L_{kd}$ where $\alpha$ is weight term.

As presented in Table 7, incorporating knowledge distillation ("GT+KD") consistently outperforms using only ground truth labels ("GT") across different approaches for MTEB benchmarks. Among the methods, 2:4 semi-structured pruning yields the worst results but benefits the most from knowledge distillation, achieving improvements of 0.76 on the MTEB benchmark.

Table 7: Knowledge Distillation - MTEB benchmark

| Label Types | Semi-structured | | Unstructured |
| | 2:4 | 4:8 | |
| --- | --- | --- | --- |
| GT | 70.41 | 70.90 | 71.18 |
| GT+KD | 71.17 | 71.22 | 71.48 |

## A.3 QUANTIZATION

For weight quantization stage, we adopt GPTQ (Frantar et al., 2022), a post-training weight quantization method that utilizes approximate Hessian information to reduce the precision of the weights. To evaluate our compressed embedding models, we compare them against three smaller LLM-based embedding models—Llama3.2-3B, Qwen2.5-3B, and Minitron-4B—which have varying numbers of weight parameters. Table 8 provides the averaged MTEB scores for compressed models (pruning and quantization), respectively.

A key observation is that our compressed models demonstrates superior robustness in low precision settings compared to their smaller counter parts. For example, NV-Embed quantized to INT8 maintains nearly identical MTEB scores (0.0% for 2:4 semi-structured, 0.01% for 4:8 semi-structured, and 0.01% for unstructured) compared to the performance drops observed in smaller models such as Llama-3B (-0.47%), Qwen-3B (-0.14%), and Minitron-4B (-0.84%). This trend remains consistent across different 8 bit precision cases as well.

Compared to integer format which has an uniform numerical distribution, floating point format can also represent the same number of discrete points, covering larger numerical range and non-uniform distributions (high precision for small values and lower precision for large values). There are two primary FP8 format: E4M3 (4-bit exponent, 3-bit mantissa), E5M2 (5-bit exponent, 2-bit mantissa) where 1 bit represents the signed bit. Table 8 shows that 8 bit floating point (E4M3 and E5M2) achieve comparable MTEB scores to the INT8 format.

Table 8: Quantization - MTEB benchmark

| | Precision | 16bit | INT8 | FP8 (E4M3) | FP8 (E5M2) |
| --- | --- | --- | --- | --- | --- |
| NV-Embed (2:4) | Score | 71.17 | 71.17 | 70.94 | 71.14 |
| | Diff (%) | - | 0.00% | -0.34% | 0.03% |
| NV-Embed (4:8) | Score | 71.22 | 71.23 | 71.28 | 71.48 |
| | Diff (%) | - | 0.01% | 0.08% | 0.37% |
| NV-Embed (Unstr) | Score | 71.48 | 71.49 | 71.55 | 71.75 |
| | Diff (%) | - | 0.01% | 0.09% | 0.37% |
| Llama3.2-3b | Score | 70.31 | 69.98 | 70.05 | 70.06 |
| | Diff (%) | - | -0.47% | -0.36% | -0.35% |
| Qwen2.5-3b | Score | 69.77 | 69.70 | 69.70 | 69.67 |
| | Diff (%) | - | -0.1% | -0.1% | -0.14% |
| Minitron-4b | Score | 70.68 | 70.09 | 69.97 | 69.97 |
| | Diff (%) | - | -0.84% | -1.0% | -1.02% |

# B  AIR BENCHMARK

In this appendix section, we present AIR-Bench[3] (version of 24.04) that is newly released information retrieval benchmark, incorporating the diverse and comprehensive domains such as healthcare, law, news, book, arxiv, finance and synthetically generated samples using diverse LLMs. Importantly, AIR-Bench can help us to understand the generalization capability of the embedding/retrieval model, because the majority of different domain samples do not appear in MTEB benchmarks. Moreover, the AIR-Bench is designed as a closed-book benchmark whose ground truth is kept confidential. As a result, the benchmark score can be only obtained through the HuggingFace Hub platform.

In AIR-Benchmark 24.04 version, there are two tasks: QA and Long-Doc. We run evaluations on 8 English datasets in QA task and 15 English datasets on the Long-Doc task. As shown in Table 9, our NV-Embed-v2 achieves the second highest scores in QA section. As described in Table 10, our NV-Embed-v2 attained the highest scores of 74.78 on the Long-Doc section, surpassing the Bge-en-icl model that requires overheads adding in-context examples to query during training. It is important to highlight that the NV-Embed-v2 model, which achieved higher MTEB accuracy scores, also demonstrates improved accuracy on both QA and Long-Doc tasks in the AIR-Bench compared to NV-Embed-v1. Interestingly, this is not always observed in the literature, where a model performing better on MTEB does not necessarily outperform on the AIR-Bench. For example, while SFR-Embedding-2R substantially outperforms SFR-Embedding-Mistral in MTEB scores (SFR-Embedding-2R: 70.31, SFR-Embedding-Mistral: 67.56), it falls short in AIR-Bench performance both in QA (SFR-Embedding-2R: 49.47, SFR-Embedding-Mistral: 51.58) and Long-doc (SFR-Embedding-2R: 67.45, SFR-Embedding-Mistral: 69.0).

Table 9: QA (nDCG@10 scores) on AIR benchmark 24.04

| Domain | Wiki | Web | News | Healthcare | Law | Finance | Arxiv | Msmarco | Avg (8) |
|---|---|---|---|---|---|---|---|---|---|
| Bge-en-icl (zero-shot) | 64.61 | 54.40 | 55.11 | 57.25 | 25.10 | 54.81 | 48.46 | 63.71 | 52.93 |
| NV-Embed-v2 | 65.19 | 52.58 | 53.13 | 59.56 | 25.00 | 53.04 | 48.94 | 60.8 | **52.28** |
| SFR-Embedding-Mistral | 63.46 | 51.27 | 52.21 | 58.76 | 23.27 | 56.94 | 47.75 | 58.99 | 51.58 |
| Stella-1.5B-v5 | 61.99 | 50.88 | 53.87 | 58.81 | 23.22 | 57.26 | 44.81 | 61.38 | 51.53 |
| Gte-Qwen2-7B-instruct | 63.46 | 51.20 | 54.07 | 54.20 | 22.31 | 58.20 | 40.27 | 58.39 | 50.26 |
| NV-Embed-v1 | 62.84 | 50.42 | 51.46 | 58.53 | 20.65 | 49.89 | 46.10 | 60.27 | **50.02** |
| Linq-Embed-Mistral | 61.04 | 48.41 | 49.44 | 60.18 | 20.34 | 50.04 | 47.56 | 60.50 | 49.69 |
| SFR-Embedding-2R | 63.72 | 48.77 | 51.14 | 55.86 | 20.98 | 54.78 | 42.84 | 57.66 | 49.47 |
| E5-mistral-7b-instruct | 61.67 | 44.41 | 48.18 | 56.32 | 19.32 | 54.79 | 44.78 | 59.03 | 48.56 |

Table 10: Long-document (Recall@10 scores) on AIR benchmark 24.04

| Domain | Arxiv (4) | Book (2) | Healthcare (5) | Law (4) | Avg. (15) |
|---|---|---|---|---|---|
| NV-Embed-v2 | 79.27 | 77.46 | 73.01 | 71.18 | **74.78** |
| Bge-en-icl (zero-shot) | 78.30 | 78.21 | 73.65 | 67.09 | 73.75 |
| NV-Embed-v1 | 77.65 | 75.49 | 72.38 | 69.55 | **73.45** |
| Bge-multilingual-gemma2 | 71.77 | 76.46 | 73.96 | 70.86 | 72.88 |
| Linq-Embed-Mistral | 75.46 | 73.81 | 71.58 | 68.58 | 72.11 |
| Stella-1.5B-v5 | 73.17 | 74.38 | 70.02 | 69.32 | 71.25 |
| SFR-Embedding-Mistral | 72.79 | 72.41 | 67.94 | 64.83 | 69.0 |
| Text-embed-3-large (OpenAI) | 74.53 | 73.16 | 65.83 | 64.47 | 68.77 |
| E5-mistral-7b-instruct | 72.14 | 72.44 | 68.44 | 62.92 | 68.49 |
| SFR-Embedding-2R | 70.51 | 70.22 | 67.60 | 62.82 | 67.45 |

---

[3]https://github.com/AIR-Bench/AIR-Bench

# C  EXPERIMENTAL DETAILS AND INSTRUCTION TEMPLATES FOR TRAINING AND EVALUATION

In this section, we describe our detailed experimental setups. We use a parameter-efficient finetuning (PEFT) method denoted as low-rank adaptation (LoRA) (Hu et al., 2021) to efficiently finetune our proposed NV-Embed model. We chose Mistral 7B (Jiang et al., 2023) as the base decoder-only LLM. We replace the attention mask from causal to bidirectional, and integrate the latent attention layer with 512 latents, 4096 hidden dimension size, and 8 multi-head attentions.

We train Mistral 7B LLM model end-to-end with a contrastive loss using LoRA with rank 16, alpha 32 and dropout rate of 0.1. We use Adam optimizer with 50 warm-up steps and learning rate 2e-5 for first stage and 1.5e-5 for second stage with linear decay. The optimizer hyperparameters are included in Table 11. We restart the optimizer with the same 50 warm-up steps and lower learning rate for the second stage. The model is finetuned with 128 batch size, where each batch is composed of a query paired with 1 positive and 7 hard negative documents. Training samples from different datasets in Table 12 are uniformly sampled. We train using Bfloat16, and set the maximum sequence length as 512 tokens. The special <BOS> and <EOS> tokens are appended at the start and end of given query and documents. The whole training is conducted in two stages where the model is initially trained on retrieval datasets utilizing in-batch negative technique. Subsequently, the model is trained with blended datasets with both retrieval and non-retrieval embedding tasks.

For evaluation, we assess our model using a maximum length of 512 tokens to ensure fair comparisons with prior work (Wang et al., 2023b), which also provides evaluation results based on 512 token limits. Evaluation instructions templates are available in Table 13.

Table 11: Parameters used in the experiments

| Parameter | Value |
|---|---|
| Batchsize | 128 |
| Number of Hardnegatives | 7 |
| Warm-up Steps | 50 |
| Training Steps | First stage - 20k
Second stage - 18k |
| Learning Rate | First stage - 2e-5
Second stage - 1.5e-5 |
| LoRA Params | Rank - 16
Alpha - 32
Dropout - 0.1 |
| Weight Decay | 0.03 |
| Optimizer | Adam |
| Padding Side | right |
| Number of Latents ($r$) | 512 |
| Latent Width ($d$) | 4096 |
| Multi-Attention Heads | 8 |

Table 12: Instructions and number of samples used for each training dataset.

| Task Name | Instruction Template | Number of Samples |
|---|---|---|
| ArguAna | Given a claim, retrieve documents that support or refute the claim | 16k |
| Natural Language Inference | Retrieve semantically similar text
Given a premise, retrieve a hypothesis that is entailed by the premise | 270k |
| PAQ, MSMARCO | Given a web search query, retrieve relevant passages that answer the query
Given a question, retrieve passages that answer the question | 500k, 500k |
| | Given a question, retrieve documents that can help answer the question | |
| SQUAD | Given a question, retrieve passages that answer the question | 87k |
| StackExchange | Given a web search query, retrieve relevant passages that answer the query | 80k |
| Natural Question | Given a question, retrieve passages that answer the question | 100k |
| HotpotQA | Given a multi-hop question, retrieve documents that can help answer the question | 170k |
| FEVER | Given a claim, retrieve documents that support or refute the claim | 140k |
| FiQA2018 | Given a financial question, retrieve relevant passages that answer the query | 5k |
| BioASQ | Given a query, retrieve documents that can help answer the question | 2.4k |
| HoVer | Given a claim, retrieve documents that support or refute the claim | 17k |
| Nfcorpus | Given a question, retrieve relevant documents that answer the question | 3.6k |
| MIRACL | Given a question, retrieve passages that answer the question | 2k |
| Mr.TyDi | Given a question, retrieve passages that answer the question | 2k |
| SciFact | Given a scientific claim, retrieve documents that support or refute the claim | 0.9k |
| STS12, STS22, STSBenchmark | Retrieve semantically similar text. | 1.8k, 0.3k, 2.7k |
| AmazonCounterfactual-Classification | Classify a given Amazon customer review text as either counterfactual or not-counterfactual | 6k |
| AmazonPolarity-Classification | Classify Amazon reviews into positive or negative sentiment | 20k |
| AmazonReviews-Classification | Classify the given Amazon review into its appropriate rating category | 40k |
| Banking77-Classification | Given a online banking query, find the corresponding intents | 10k |
| Emotion-Classification | Classify the emotion expressed in the given Twitter message into one of the six emotions:anger, fear, joy, love, sadness, and surprise | 16k |
| Imdb-Classification | Classify the sentiment expressed in the given movie review text from the IMDB dataset | 24k |
| MTOPIntent-Classification | Classify the intent of the given utterance in task-oriented conversation | 15k |
| MTOPDomain-Classification | Classify the intent domain of the given utterance in task-oriented conversation | 15k |
| MassiveIntent-Classification | Given a user utterance as query, find the user intents | 11k |
| MassiveScenario-Classification | Given a user utterance as query, find the user scenarios | 11k |
| ToxicConversationsClassification | Classify the given comments as either toxic or not toxic | 50k |
| TweetSentimentExtractionClassification | Classify the sentiment of a given tweet as either positive, negative, or neutral | 27k |
| Arxiv-Clustering-P2P | Identify the main and secondary category of Arxiv papers based on the titles and abstracts | 50k |
| Arxiv-Clustering-S2S | Identify the main and secondary category of Arxiv papers based on the titles | 50k |
| Biorxiv-Clustering-P2P | Identify the main category of Biorxiv papers based on the titles and abstracts | 15k |
| Biorxiv-Clustering-S2S | Identify the main category of Biorxiv papers based on the titles | 15k |
| Medrxiv-Clustering-P2P | Identify the main category of Medrxiv papers based on the titles and abstracts | 2.3k |
| Medrxiv-Clustering-S2S | Identify the main category of Medrxiv papers based on the titles | 2.3k |
| Reddit-Clustering | Identify the main category of Medrxiv papers based on the titles and abstracts | 50k |
| Reddit-Clustering-S2S | Identify the main category of Medrxiv papers based on the titles | 40k |
| Stackexchange-Clustering | Identify the main category of Medrxiv papers based on the titles and abstracts | 50k |
| Stackexchange-Clustering-S2S | Identify the main category of Medrxiv papers based on the titles | 40k |
| TwentyNewsgroups-Clustering | Identify the topic or theme of the given news articles | 1.7k |

# D LATENT-ATTENTION VISUALIZATION

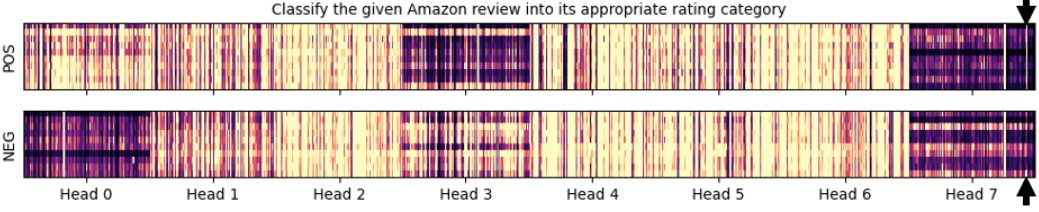

**Latent attention over AmazonReviewsClassification reviews**

Classify the given Amazon review into its appropriate rating category

Figure 2: Attention over 4096 latents across 8 heads (columns) are visualized for 10 positive and 10 negative reviews (rows) from the AmazonReviewsClassification dataset. The attention weights are mean pooled across tokens. The attention weights reveal that the latents specialize in learning features of queries. The latent indicated by the arrows specialized in learning the positivity of reviews. It has high attention across the positive reviews and low attention across the negative reviews.

Table 13: Instructions used for evaluation on the MTEB benchmark. "STS*" indicates we use the same instructions for all the STS tasks.

| Task Name | Instruction Template |
|---|---|
| ArguAna | Given a claim, retrieve documents that support or refute the claim |
| ClimateFEVER | Given a claim about climate change, retrieve documents that support or refute the claim |
| DBPedia | Given a query, retrieve relevant entity descriptions from DBPedia |
| FEVER | Given a claim, retrieve documents that support or refute the claim |
| FiQA2018 | Given a financial question, retrieve user replies that best answer the question |
| HotpotQA | Given a multi-hop question, retrieve documents that can help answer the question |
| MSMARCO | Given a web search query, retrieve relevant passages that answer the query |
| NFCorpus | Given a question, retrieve relevant documents that answer the question |
| Natural Question | Given a question, retrieve passages that answer the question |
| QuoraRetrieval | Given a question, retrieve questions that are semantically equivalent to the given question |
| SCIDOCS | Given a scientific paper title, retrieve paper abstracts that are cited by the given paper |
| SciFact | Given a scientific claim, retrieve documents that support or refute the claim |
| Touche2020 | Given a question, retrieve passages that answer the question |
| TREC-COVID | Given a query on COVID-19, retrieve documents that answer the query |
| STS | Retrieve semantically similar text. |
| SummEval | Given a news summary, retrieve other semantically similar summaries |
| AmazonCounterfactualClassification | Classify a given Amazon customer review text as either counterfactual or not-counterfactual |
| AmazonPolarityClassification | Classify Amazon reviews into positive or negative sentiment |
| AmazonReviewsClassification | Classify the given Amazon review into its appropriate rating category |
| Banking77Classification | Given a online banking query, find the corresponding intents |
| EmotionClassification | Classify the emotion expressed in the given Twitter message into one of the six emotions:anger, fear, joy, love, sadness, and surprise |
| ImdbClassification | Classify the sentiment expressed in the given movie review text from the IMDB dataset |
| MassiveIntentClassification | Given a user utterance as query, find the user intents |
| MassiveScenarioClassification | Given a user utterance as query, find the user scenarios |
| MTOPDomainClassification | Classify the intent domain of the given utterance in task-oriented conversation |
| MTOPIntentClassification | Classify the intent of the given utterance in task-oriented conversation |
| ToxicConversationsClassification | Classify the given comments as either toxic or not toxic |
| TweetSentimentExtractionClassification | Classify the sentiment of a given tweet as either positive, negative, or neutral |
| ArxivClusteringP2P | Identify the main and secondary category of Arxiv papers based on the titles and abstracts |
| ArxivClusteringS2S | Identify the main and secondary category of Arxiv papers based on the titles |
| BiorxivClusteringP2P | Identify the main category of Biorxiv papers based on the titles and abstracts |
| BiorxivClusteringS2S | Identify the main category of Biorxiv papers based on the titles |
| MedrxivClusteringP2P | Identify the main category of Medrxiv papers based on the titles and abstracts |
| MedrxivClusteringS2S | Identify the main category of Medrxiv papers based on the titles |
| RedditClustering | Identify the topic or theme of Reddit posts based on the titles |
| RedditClusteringP2P | Identify the topic or theme of Reddit posts based on the titles and posts |
| StackExchangeClustering | Identify the topic or theme of StackExchange posts based on the titles |
| StackExchangeClusteringP2P | Identify the topic or theme of StackExchange posts based on the given paragraphs |
| TwentyNewsgroupsClustering | Identify the topic or theme of the given news articles |
| AskUbuntuDupQuestions | Retrieve duplicate questions from AskUbuntu forum |
| MindSmallReranking | Retrieve relevant news articles based on user browsing history |
| SciDocsRR | Given a title of a scientific paper, retrieve the titles of other relevant papers |
| StackOverflowDupQuestions | Retrieve duplicate questions from StackOverflow forum |
| SprintDuplicateQuestions | Retrieve duplicate questions from Sprint forum |
| TwitterSemEval2015 | Retrieve tweets that are semantically similar to the given tweet |
| TwitterURLCorpus | Retrieve tweets that are semantically similar to the given tweet |

Table 14: Full BEIR and MTEB benchmark

| | Bge-multilingual-gemma2 | Gte-Qwen2-7B-instruct | SFR-Embedding-2R | Stella-en-1.5B-v5 | bge-en-icl (zeroshot) | NV-Embed-v1 | NV-Embed-v2 |
|---|---|---|---|---|---|---|---|
| ArguAna | 77.37 | 64.27 | 62.34 | 65.27 | 82.76 | 68.21 | 70.07 |
| ClimateFEVER | 39.37 | 45.88 | 34.43 | 46.11 | 45.35 | 34.72 | 45.39 |
| CQADupStack | 47.94 | 46.43 | 46.11 | 47.75 | 47.23 | 50.51 | 50.24 |
| DBPEDIA | 51.37 | 52.42 | 51.21 | 52.28 | 50.42 | 48.29 | 53.50 |
| FEVER | 90.38 | 95.11 | 92.16 | 94.83 | 91.96 | 87.77 | 93.75 |
| FiQA2018 | 60.04 | 62.03 | 61.77 | 60.48 | 58.77 | 63.1 | 65.73 |
| HotpotQA | 83.26 | 73.08 | 81.36 | 76.67 | 84.98 | 79.92 | 85.48 |
| MSMARCO | 45.71 | 45.98 | 42.18 | 45.22 | 46.72 | 46.49 | 45.63 |
| NFCorpus | 38.11 | 40.6 | 41.34 | 42 | 40.69 | 38.04 | 45.17 |
| Natural | 71.45 | 67 | 73.96 | 71.8 | 73.85 | 71.22 | 73.57 |
| QuoraRetrieval | 90.04 | 90.09 | 89.58 | 90.03 | 91.02 | 89.21 | 89.04 |
| SCIDOCS | 26.93 | 28.91 | 24.87 | 26.64 | 25.25 | 20.19 | 21.90 |
| SciFact | 72.05 | 79.06 | 85.91 | 80.09 | 78.33 | 78.43 | 80.13 |
| Touche2020 | 30.26 | 30.57 | 28.18 | 29.94 | 29.67 | 28.38 | 31.78 |
| TREC-COVID | 64.27 | 82.26 | 87.28 | 85.98 | 78.11 | 85.88 | 88.44 |
| BIOSSES | 85.74 | 81.37 | 87.6 | 83.11 | 86.35 | 85.59 | 87.42 |
| SICK-R | 82.66 | 79.28 | 77.01 | 82.89 | 83.87 | 82.8 | 82.15 |
| STS12 | 77.71 | 79.55 | 75.67 | 80.09 | 77.73 | 76.22 | 77.89 |
| STS13 | 87.45 | 88.83 | 82.4 | 89.68 | 85.98 | 86.3 | 88.30 |
| STS14 | 83.48 | 83.87 | 79.93 | 85.07 | 82.34 | 82.09 | 84.30 |
| STS15 | 87.63 | 88.54 | 85.82 | 89.39 | 87.35 | 87.24 | 89.04 |
| STS16 | 86.7 | 86.49 | 84.5 | 87.15 | 86.54 | 84.77 | 86.77 |
| STS17 | 91.18 | 88.73 | 88.93 | 91.35 | 91.25 | 87.42 | 90.67 |
| STS22 | 69.02 | 66.88 | 67.1 | 68.1 | 68.08 | 69.85 | 68.12 |
| STSBenchmark | 87.25 | 86.85 | 83.6 | 88.23 | 87.92 | 86.14 | 88.41 |
| SummEval | 31.2 | 31.35 | 30.71 | 31.49 | 30.75 | 31.2 | 30.70 |
| SprintDuplicateQuestions | 90.94 | 92.82 | 97.62 | 96.04 | 95.06 | 95.94 | 97.02 |
| TwitterSemEval2015 | 79.64 | 77.96 | 78.57 | 80.58 | 78.54 | 78.73 | 81.11 |
| TwitterURLCorpus | 86.95 | 86.59 | 88.03 | 87.58 | 87.19 | 86.05 | 87.87 |
| AmazonCounterfactual | 89.48 | 91.31 | 92.72 | 92.87 | 92.88 | 95.12 | 94.28 |
| AmazonPolarity | 96.9 | 97.5 | 97.31 | 97.16 | 96.86 | 97.14 | 97.74 |
| AmazonReviews | 61.6 | 62.56 | 61.04 | 59.36 | 61.28 | 55.47 | 63.96 |
| Banking77 | 92.53 | 87.57 | 90.02 | 89.79 | 91.42 | 90.34 | 92.42 |
| Emotion | 92.97 | 79.45 | 93.37 | 84.29 | 93.31 | 91.71 | 93.38 |
| Imdb | 96.66 | 96.75 | 96.8 | 96.66 | 96.91 | 97.06 | 97.14 |
| MassiveIntent | 82.05 | 85.41 | 85.97 | 85.83 | 82.26 | 80.07 | 86.10 |
| MassiveScenario | 84.4 | 89.77 | 90.61 | 90.2 | 83.92 | 81.74 | 92.17 |
| MTOPDomain | 98.61 | 99.04 | 98.58 | 99.01 | 97.99 | 96.51 | 99.25 |
| MTOPIntent | 95.51 | 91.88 | 91.3 | 92.78 | 93.56 | 89.77 | 94.37 |
| ToxicConversations | 87.34 | 85.12 | 91.14 | 88.76 | 93.16 | 92.6 | 92.74 |
| TweetSentimentExtraction | 78.86 | 72.58 | 79.7 | 74.84 | 79.9 | 80.6 | 80.87 |
| Arxiv-P2P | 54.91 | 54.46 | 54.02 | 55.44 | 54.42 | 53.76 | 55.80 |
| Arxiv-S2S | 50.28 | 51.74 | 48.82 | 50.66 | 49.17 | 49.59 | 51.26 |
| Biorxiv-P2P | 52.64 | 50.09 | 50.76 | 50.68 | 52.32 | 48.15 | 54.09 |
| Biorxiv-S2S | 49.2 | 46.65 | 46.57 | 46.87 | 48.38 | 44.74 | 49.60 |
| Medrxiv-P2P | 45.81 | 46.23 | 46.66 | 46.87 | 46.13 | 39.24 | 46.09 |
| Medrxiv-S2S | 44.11 | 44.13 | 44.18 | 44.65 | 44.2 | 36.98 | 44.86 |
| Reddit | 56.03 | 73.55 | 62.92 | 72.86 | 71.2 | 63.2 | 71.10 |
| Reddit-P2P | 65.83 | 74.13 | 72.74 | 75.27 | 72.17 | 68.01 | 74.94 |
| StackExchange | 66.21 | 79.86 | 76.48 | 80.29 | 81.29 | 74.99 | 82.10 |
| StackExchange-P2P | 45.74 | 49.41 | 48.29 | 49.57 | 45.53 | 42.04 | 48.36 |
| TwentyNewsgroups | 70.44 | 53.91 | 66.42 | 61.43 | 68.51 | 60.13 | 64.82 |
| AskUbuntuDupQuestions | 64.59 | 67.58 | 66.71 | 67.33 | 64.8 | 67.5 | 67.46 |
| MindSmallRerank | 31.79 | 33.36 | 31.26 | 33.05 | 30.6 | 30.82 | 31.76 |
| SciDocsRR | 87.6 | 89.09 | 87.29 | 89.2 | 86.9 | 87.26 | 87.59 |
| StackOverflowDupQuestions | 54.9 | 55.66 | 55.32 | 55.25 | 56.32 | 56.58 | 55.79 |
| **MTEB Average (56)** | 69.88 | 70.24 | 70.31 | 71.19 | 71.24 | 69.32 | **72.31** |

Table 15: Prompt template for short-long matching subgroup.

Brainstorm a list of potentially useful text retrieval tasks.

Here are a few examples for your reference:
- Given a web search query, retrieve relevant passages that answer the query
- Given a claim about climate change, retrieve documents that support or refute the claim
- Given a job title, search for job descriptions that provide information about the role

Please adhere to the following guidelines:
- Specify the type of query and the type of desired texts.
- Each retrieval task should cover a wide range of queries, and should not be too specific.
- Cover a wide range of query types and desired text types.

Your output must always be a JSON list of strings only, with about 40 elements, and each element corresponds to
a distinct retrieval task in one sentence. Do not explain yourself or output anything else. Be creative!

You have been assigned a retrieval task: {task}

Your mission is to write one text retrieval example for this task in JSON format. The JSON object must
contain the following keys:
- "user_query": a string, a random example of what is provided as specified by the task description.
- "positive_document": a string, a relevant document for the user query.
- "hard_negative_document1": a string, a hard negative document that is irrelevant but appears relevant to the query.
- "hard_negative_document2": a string, another hard negative document that is irrelevant but appears relevant to the query.

Please adhere to the following guidelines:
- The "user_query" should be {query_type}, {query_length}, {clarity}, and diverse in topic. The "user_query" should
not restate the task and just contain what the task description says is provided.
- All documents must be created independent of the query. Avoid copying the query verbatim. It's acceptable if
some parts of the "positive_document" are not topically related to the query.
- All documents should be at least {num_words} words long.
- The "hard_negative_document1" may contain little useful information, but it should be less useful or
comprehensive compared to the "positive_document".
- The "hard_negative_document2" may should be about a related but different topic.
- Do not provide any explanation in any document on why it is relevant or not relevant to the query.
- Both the query and documents require {difficulty} level education to understand.

Your output must always be a JSON object only, do not explain yourself or output anything else. Be creative!"""

Placeholders:
"{query_type}" ∈ {extremely long-tail, long-tail, common}
"{query_length}" ∈ {less than 5 words, 5 to 15 words, at least 10 words}
"{difficulty}" ∈ {high school, college, PhD}
"{clarity}" ∈ {clear, understandable with some effort, ambiguous}
"{num_words}" ∈ {50, 100, 200, 300, 400, 500}

Table 16: Prompt template for long-short matching subgroup.

---

Brainstorm a list of potentially useful text classification tasks.

Please adhere to the following guidelines:
- Tasks should cover a diverse range of domains and task types.

Your output must always be a JSON list of strings only, with about 40 elements, and each element corresponds to a distinct text classification task in one sentence. Do not explain yourself or output anything else. Be creative!

---

You have been assigned a text classification task: {task}

Your mission is to write one text classification example for this task in JSON format. The JSON object must contain the following keys:
- "input_text": a string, the input text specified by the classification task.
- "label": a string, the correct
label of the input text.
- "misleading_label": a string, an incorrect label that is related to the task.

Please adhere to the following guidelines:
- The "input_text" should be {num_words} words and diverse in expression.
- The "misleading_label" must be a valid label for the given task, but not as appropriate as the "label" for the "input_text".
- Avoid including the values of the "label" and "misleading_label" fields in the "input_text", that would make the task too easy.
- The "input_text" is {clarity} and requires {difficulty} level education to comprehend.

Your output must always be a JSON object only, do not explain yourself or output anything else. Be creative!

---

Placeholders:
{num_words} ∈ {"less than 10","at least 10", "at least 50", "at least 100", "at least 200"}
{difficulty} ∈ {high school, college, PhD}
{clarity} ∈ {clear, understandable with some effort, ambiguous}

---

