# OpenReview forum: "NV-Embed: Improved Techniques for Training LLMs as Generalist Embedding Models"
_ICLR.cc/2025/Conference — ICLR 2025 Spotlight_

### Official Review · Reviewer_wm7z · 2024-11-02

**Soundness:** 3
**Presentation:** 3
**Contribution:** 3
**Rating:** 8
**Confidence:** 3

**Summary:**

This paper presents the NV-Embed-v1 and v2 systems for general purpose text embeddings. The paper presents the learning algorithm along with architecture variations (latent attention), ywhich describes how the system achieves #1 on the MTEB leaderboard at the time of submission. This paper documents the high performing system, which is likely to have gathered much attention given its position in the leaderboard.

**Strengths:**

This paper severs a very clear and important purpose -- documenting the current top performing system on the extremely competitive text embedding benchmark, MTEB. The paper presents an approach, which like other approaches on the MTEB leaderboard, takes a pretrained transformer model and trains using a mixture of datasets/tasks, including a hard negative mining pipeline. The authors also introduce a simple architectural change, latent attention layer.

The key strength of the paper, is its empirical gains and empirical analysis. The authors provide benchmark performance on a wide range of settings with highly performing models. This shows the capabilities of models on a wide variety of tasks, including important ablations regarding attention type, stages of training, etc.

**Weaknesses:**

As this paper is primarily the documentation of a very highly performing empirical system, main weakness I would point out is about innovation. On one level the paper is ground breaking because of its empirical gains, on another, the core methodological techniques are well known, only that they are more effectively performed and analyzed here.

The latent attention mechanism, while a nice architectural change, only has a small effect on the average performance (Table 2). The hard negative mining pipeline, a more established technique, on the other hand makes much more of a difference (Table 4).

**Questions:**

For each stage of training, do you use the same optimizer? Restarting it for each stage?

---

> ### Author Response · Authors · 2024-11-22
>
> Thank you for summarizing our paper nicely and highlighting the strengths. We discuss your raised points in the following.
>
> > “As this paper is primarily the documentation of a very highly performing empirical system, main weakness I would point out is about innovation. On one level the paper is ground breaking because of its empirical gains, on another, the core methodological techniques are well known, only that they are more effectively performed and analyzed here..”
> - We agree that NV-Embed is a combination of novel (e.g., latent attention, two staged training, example-baed labeling technique) and established techniques (e.g., hard negative mining, synthetic sample generation), and we appreciate your acknowledgement that this is ground breaking because of its empirical gains. We believe this reflects the nature of the problem itself: achieving state-of-the-art performance on the MTEB benchmark, which spans 56 diverse tasks, necessitates the integration of the best techniques available. Importantly, the challenge of combining these techniques to achieve significant improvements is substantial and requires non-trivial efforts.
>
>
> > “The latent attention mechanism, while a nice architectural change, only has a small effect on the average performance (Table 2). The hard negative mining pipeline, a more established technique, on the other hand makes much more of a difference (Table 4).”
> - Thank you for asking this question.  We would like to contextualize that the score gaps of leading models on the MTEB benchmark (w/ 56 diverse tasks) are narrow, usually below 0.5. During the period of NV-Embed-{v1, v2} release, six other models have been released in MTEB leaderboard as noted below table, each contributing improvements of 0.1 to 0.8 points in MTEB averaged scores. The  overall MTEB score improvement from latent attention over mean pooling is 71.71 to 72.31 (+0.6 for NV-Embed-v2), which are therefore non-marginal relative to the improvements observed between competing approaches.
>
> | MTEB Rank |           Name          |  MTEB score | Released date |
> |:---------:|:-----------------------:|:-----------------------------:|:-------------:|
> |     1     |        NV-Embed-v2       |   72.31   | Aug 30, 2024 |
> |     2     |        Bge-en-icl       |     71.67   | July 25, 2024 |
> |     3     |    stella_en_1.5B_v5    |       71.19   | July 12, 2024 |
> |     4     |    SFR-Embedding-2_R    |     70.31   | June 18, 2024 |
> |     5     |  gte-Qwen2-7B-instruct  |   70.24   | June 17, 2024 |
> |     6     |    stella_en_400M_v5    |        70.11   | July 12, 2024 |
> |     7     | bge-multilingual-gemma2 |     69.88   | July 25, 2024 |
> |     8     |       NV-Embed-v1       |       69.32   |  May 24, 2024 |
> |     9     | voyage-large-2-instruct |     68.23   |  May 5, 2024   |
>
> - Our proposed techniques of model architecture (latent attention), training approach (two-stage instruction tuning), and data curation strategy (hard negative mining, synthetic dataset generation, and example-based multi-class labeling) synergize with each other, to achieve state-of-the-art embedding models. We acknowledge that the hard negative mining technique contributes (+ 1.1 overall MTEB scores from [S0] to [S1] in Table 4 of manuscript) to performance improvements. Additionally, the proposed two-stage training strategy has improved both retrieval (+1.3) and overall accuracy (+0.54 overall MTEB score, as shown in Table 4). Furthermore, the example-based labeling approach has led to gains, particularly in clustering (+5.4) and overall accuracy (+1.5 overall MTEB score). We observe the relatively higher score gains for data curation strategy, but our model architecture and training approach also provides significant and non-trivial benefits. As a result, our approach of integrating novel architectural designs with well-established optimization strategies to achieve superior performance across a variety of embedding tasks. We hope this explanation highlights the complementary strengths of these novel contributions.
>
> > “For each stage of training, do you use the same optimizer? Restarting it for each stage?”
> - Thank you for the question. We use the Adam optimizer for each training stage. The optimizer hyperparameters are included in Table 8.  We restart the optimizer with the same 50 warm-up steps and lower learning rate for the second stage. We have updated the paper draft to clarify this.

---

> > ### Comment · Reviewer_wm7z · 2024-11-27
> >
> > Thank you for your detailed clarifications and response. This is very helpful.

---

### Official Review · Reviewer_9J2v · 2024-11-04

**Soundness:** 2
**Presentation:** 3
**Contribution:** 3
**Rating:** 6
**Confidence:** 4

**Summary:**

This paper proposes a new embedding model: NV-embedder. The model uses a latent attention layer to obtain pooled embeddings instead of the common EOS token. The authors also propose a two-stage training method and employ hard example mining and data synthesis.

**Strengths:**

- The authors tested the model on a leaderboard outside of MTEB. Since the training data is related to MTEB, out-of-domain testing is necessary.
- The authors conducted extensive ablation studies to verify the effectiveness of each module.

**Weaknesses:**

The biggest drawback is the complexity of the entire training process, which includes multi-stage training, hard example mining, and data synthesis. This involves too many operations, making it difficult to reproduce.

**Questions:**

no

---

> ### Author Response · Authors · 2024-11-22
>
> Many thanks for your comments and feedback. We discuss your raised point in the following.
>
> > “The biggest drawback is the complexity of the entire training process, which includes multi-stage training, hard example mining, and data synthesis. This involves too many operations, making it difficult to reproduce.”
> - Complexity: Thanks for the comment. We made every effort to keep our method straightforward and ensure the presentation is clear. As you mentioned, we conduct extensive ablation studies to validate the effectiveness of each module: model architecture (e.g., latent attention), training method (e.g., two-stage instruction tuning), and data curation strategies (e.g., hard-negative mining, synthetic dataset generation, and example-based multi-class labeling). We believe the resulting complexity of training process really reflects the nature of the problem itself: achieving state-of-the-art performance on the MTEB benchmark, which spans 56 diverse tasks, requires integrating several techniques.
>
> - Reproducibility: In the paper, we include abundant technical details for reproduction, including a detailed training recipe with all hyperparameters in Table 8 and training data blend information in Table 9. We also release the model weights and provide instructions for the reproduction of evaluation results in Table 10.

---

### Official Review · Reviewer_rURm · 2024-11-04

**Soundness:** 3
**Presentation:** 2
**Contribution:** 2
**Rating:** 8
**Confidence:** 3

**Summary:**

This paper presents techniques for leveraging pretrained decoder-only large transformers in retrieval tasks, achieving state-of-the-art results on the standard retrieval benchmark (MTEB). The core methods include:
* A latent attention layer for creating pooled embeddings, which surpasses traditional mean pooling and last-token embedding approaches.
* A two-stage training process: the first stage focuses on retrieval datasets, while the second integrates non-retrieval tasks for broader versatility.
* The use of curated datasets (e.g. hard-negative mining) to further refine embedding quality.

**Strengths:**

* Provides very strong empirical results. This work achieves state-of-the-art results, securing the #1 position on the MTEB benchmark. This demonstrates the effectiveness of the proposed approach.
* Provides nice ablation studies (e.g., Table 4) to analyze the effects of various design choices, including single-stage vs. two-stage, hard negative mining (HN), public retrieval set (AD), and synthetic data generation (SD).
* Provides detailed descriptions of the data curation process, including specific techniques like hard negative mining and synthetic data generation. This transparency allows for better understanding and potential reproducibility of the work.

**Weaknesses:**

* Using decoder-only pretrained models for retrieval is not entirely novel (e.g., as seen in GritLM [1] and discussed in Section 2.2). While the authors note differences in their specific training approach, this limits the novelty of this aspect.
* Given this, the main contributions center on the latent attention layer and two-stage instruction tuning. However, the performance improvement from latent attention over mean pooling appears modest (see Table 2, bidirectional columns), raising questions about the added complexity for minimal gains.
* Although the authors provide empirical support for the benefits of two-stage training, there is limited explanation or intuition behind why this approach works effectively.

[1] Muennighoff, Niklas, et al. "Generative representational instruction tuning." arXiv preprint arXiv:2402.09906 (2024).

**Questions:**

* What would happen if the order of the two-stage training were reversed?
* Could you expand on the points raised in the weaknesses section, offering more discussion on these limitations?

---

> ### Author Response · Authors · 2024-11-22
>
> Thank you for summarizing our paper nicely and highlighting the strengths. We find all comments constructive and have tried our best to address each of them in this rebuttal.
>
> [part 1/2]
> > “Using decoder-only pretrained models for retrieval is not entirely novel (e.g., as seen in GritLM [1] and discussed in Section 2.2). While the authors note differences in their specific training approach, this limits the novelty of this aspect.”
> - Thank you for your comment. This work introduces innovations in model architecture (e.g., latent attention), training method (e.g., two-stage instruction tuning), and data curation strategies (e.g., hard-negative mining, synthetic dataset generation, and example-based multi-class labeling). By combining these techniques, we have optimized decoder-only LLMs to achieve state-of-the-art results. The NV-Embed-{v1, v2} series has not only secured and maintained the No. 1 ranking on the MTEB leaderboard but also demonstrated superior accuracy in out-of-domain tasks on the AIR Benchmark. We hope the sustained effectiveness of NV-Embed underscores the significance of our proposed methods in advancing text embedding performance in this rapidly evolving field.
>
> > “Given this, the main contributions center on the latent attention layer and two-stage instruction tuning. However, the performance improvement from latent attention over mean pooling appears modest (see Table 2, bidirectional columns), raising questions about the added complexity for minimal gains.”
> - Thank you for asking this question. We would like to contextualize that the score gaps of leading models on the MTEB benchmark (w/ 56 diverse tasks) are narrow, usually below 0.5. During the period of NV-Embed-{v1, v2} release, six other models have been released in MTEB leaderboard as noted below table, each contributing improvements of 0.1 to 0.8 points in MTEB averaged scores. The performance improvement from latent attention over mean pooling is 68.97 to 69.32 (+ 0.34 for NV-Embed-v1)  and 71.71 to 72.31 (+0.6 for NV-Embed-v2), which are therefore significant and non-trivial relative to the improvements observed between competing approaches.
>
> | MTEB Rank |           Name          |  MTEB score | Released date |
> |:---------:|:-----------------------:|:-----------------------------:|:-------------:|
> |     1     |        NV-Embed-v2       |   72.31   | Aug 30, 2024 |
> |     2     |        Bge-en-icl       |     71.67   | July 25, 2024 |
> |     3     |    stella_en_1.5B_v5    |       71.19   | July 12, 2024 |
> |     4     |    SFR-Embedding-2_R    |     70.31   | June 18, 2024 |
> |     5     |  gte-Qwen2-7B-instruct  |   70.24   | June 17, 2024 |
> |     6     |    stella_en_400M_v5    |        70.11   | July 12, 2024 |
> |     7     | bge-multilingual-gemma2 |     69.88   | July 25, 2024 |
> |     8     |       NV-Embed-v1       |       69.32   |  May 24, 2024 |
> |     9     | voyage-large-2-instruct |     68.23   |  May 5, 2024   |
>
> - Moreover, adding more parameters and computation can improve training capacity, but does not necessarily lead to improvement on test accuracy. For example, an even simpler way of adding a self-attention layer before mean pooling does not improve performance over mean pooling (results are in Table 3 and 4 of our manuscript). In contrast, the latent attention layer transforms the unrestricted representation from the transformer model to a vector space that has the bank of latent vectors as basis vectors. We hypothesize that this has a dictionary learning effect where the latent vectors learn useful sparse representations to construct the embedding.
>
> > “Although the authors provide empirical support for the benefits of two-stage training, there is limited explanation or intuition behind why this approach works effectively.”
> - Thank you for asking this question. In principle, the retrieval task presents greater difficulty compared to the other embedding tasks (such as Classification, Clustering, STS, etc), so our training strategy initially focuses on fine-tuning the model for retrieval in the first stage. Also, the two-stage training has an efficiency advantage, as the MTEB benchmark has various tasks across task types and domains, it is computationally expensive to evaluate all the tasks when iterating on experiments and to optimize data blends for performance across all tasks. With two-stage training, we could independently optimize the first stage solely for the retrieval tasks. After obtaining the best retrieval model, we could then fine-tune it in the second stage to achieve good performance in the other tasks by blending the remaining embedding tasks into the instruction-tuning.

---

> > ### Author Response · Authors · 2024-11-22
> >
> > [part 2/2]
> >
> > > “What would happen if the order of the two-stage training were reversed?.”
> > - Thank you for the insightful suggestion. We have followed your suggestion and conducted a reversed order of two-stage training during this rebuttal period. Specifically, we further finetune the “Single Stage (Inbatch disabled)” model using only the retrieval datasets with enabling the inbatch-negative technique.
> > - The MTEB scores when the order of two-stage training is reversed are presented in the last row of below table. While the retrieval score increased from 61.37 to 61.91 after the reversed two-staged training, it remains lower than the retrieval score of 62.65 achieved with our proposed two-stage training method. Furthermore, the scores on other embedding tasks, such as Clustering and STS, declined compared to the Single Stage (Inbatch disabled) approach. Consequently, the overall MTEB score for Reversed Two Stage Training (score: 71.85) is lower than our proposed Two-Stage Training (score: 72.31) as well as the Single Stage with Inbatch disabled (score: 71.94). We have incorporated these additional ablation study results into Table 4 and section 5.3.1 of the updated manuscript.
> >
> > |          Embedding Task         | Retrieval | Rerank | Cluster | PairClass | Class |  STS  | Summ. |  Avg. |
> > |:-------------------------------:|:---------:|:------:|:-------:|:---------:|:-----:|:-----:|:-----:|:-----:|
> > |  Single Stage (Inbatch enabled) |   61.25   |  60.64 |  57.67  |   87.82   |  86.6 |  83.7 | 30.75 | 70.83 |
> > | Single Stage (Inbatch disabled) |   61.37   |  60.81 |  58.31  |    88.3   |  90.2 |  84.5 | 30.96 | 71.94 |
> > |      Two Stage Training       |   62.65   |  60.65 |  58.46  |   88.67   | 90.37 | 84.31 |  30.7 | 72.31 |
> > |     Reversed Two Stage Training    |   61.91   |  60.98 |  58.22  |   88.59   | 90.26 | 83.07 | 31.28 | 71.85 |
> >
> > > “Could you expand on the points raised in the weaknesses section, offering more discussion on these limitations?”
> > - In our previous response, we include the discussions with respect to the raised points.  We have also updated our manuscript in section 3.3, 5.3.1, Table 4, and conclusion to incorporate these discussions.
> >
> > We hope our response addresses your concerns. Please let us know if you have any further questions.

---

> > > ### Comment · Reviewer_rURm · 2024-11-24
> > >
> > > Thank you to the authors for the detailed response, and especially for conducting the additional experiment (reversing the two stages). I hope this ablation can be included in the revised version, either in the main text or the appendix. I am increasing my score.

---

> > > > ### Author Response · Authors · 2024-11-26
> > > >
> > > > Dear Reviewer,
> > > >
> > > > Thank you again for your detailed comments and constructive suggestions. We will incorporate all of them into the final version of the paper.

---

### Official Review · Reviewer_MUEH · 2024-11-08

**Soundness:** 3
**Presentation:** 3
**Contribution:** 3
**Rating:** 8
**Confidence:** 5

**Summary:**

This paper gives a summary of the NV-Embed model that achieved the top performance in the MTEB benchmark.
The techniques used are
1. latent attention layer that achieves better pooling/combination of the last layer embeddings. Causal attention mask is removed during contrastive learning.
2. a two-stage contrastive instruction tuning method. First step tuning with in-batch negative and hard negative on retrieval datasets, and the second step tuning on non-retrieval datasets.
3. large amount of efforts on training data curation.

**Strengths:**

1. This work gives a good summary of the STOA NV-Embed model that leads the MTEB benchmark. I think the community will very much appreciate this paper.
2. Impressive experimental results with good ablations to justify the design choices.
3. Clear presentation.

**Weaknesses:**

The NV-Embed model is a result of a combination of methods/tricks/datasets etc.
There does not seem to be single innovative algorithm piece. This by itself is not a weakness of the work. However, as a result, the technical depth of the work is limited.

**Questions:**

The v2 model largely outperformed v1:
"We then further improve the model through the curation
of training dataset, including adding more retrieval datasets, applying positive-aware hard-negative
mining technique, using synthetic data generation process and constructing example-based multi-class
labels."
It would be nice to have more discussions and ablations specifically regarding the v2 vs. v1.

---

> ### Author Response · Authors · 2024-11-22
>
> Thank you so much for your review. We will address your comments in the following.
>
> > “The NV-Embed model is a result of a combination of methods/tricks/datasets etc. There does not seem to be single innovative algorithm piece. This by itself is not a weakness of the work. However, as a result, the technical depth of the work is limited.”
> - We agree that NV-Embed is indeed a combination of novel methods and techniques, and we appreciate your acknowledgment that this is not a weakness of the work. We believe this reflects the nature of the problem itself: achieving state-of-the-art performance on the MTEB benchmark, which spans 56 diverse tasks, requires integrating a combination of methods/tricks/datasets. Importantly, the challenge of combining these methods to achieve significant improvements is substantial and requires non-trivial technical depth.
>
> > “The v2 model largely outperformed v1: "We then further improve the model through the curation of training dataset, including adding more retrieval datasets, applying positive-aware hard-negative mining technique, using synthetic data generation process and constructing example-based multi-class labels." It would be nice to have more discussions and ablations specifically regarding the v2 vs. v1.”
> - Thank you so much for raising this question. In Table 4, we present a series of ablation studies comparing NV-Emebed-v2 and NV-Emebed-v1. We systematically ablate techniques introduced in v2 that were not applied to v1 during stage-one training, including hard-negative mining (HN), synthetically generated datasets (SD), and additional public retrieval datasets (AD). For stage-two training, we also ablate v2’s unique example-based approach, which was not used in v1. In summary, the v1 model configuration omits HN, AD, and SD in stage-one training and uses a label-based approach in stage-two training. We have updated the draft to clarify these details.

---

### Meta-Review · Area_Chair_3MZK · 2024-12-16

**Metareview:**

The paper presents a suite of techniques to improve the general purpose embeddings produced by large decoder-only LLMs. This includes a latent attention layer to create the summary embedding, a contrastive training objective for instruction tuning, and careful creation of the training data. These techniques are proven to be empirically effective, with the resulting method securing the first place in the MTEB benchmark. The paper also presents a series of ablations to identify the importance of the individual components of their method.

Building versatile and powerful text embeddings is of wide interest and import. Reviewers were unanimously favourable in their view of the paper, citing its strong empirical results and ablations as being of considerable interest to the community. The one lacuna is that the technical novelty of the individual ingredients may be somewhat limited. However, as the solid empirical results (with technical details) are expected to be useful for most future work in this space, this is not seen as diminishing the value of the work.

**Additional Comments On Reviewer Discussion:**

Initial reviews were unanimously positive. There were a few questions and suggestions regarding the experiments which the authors adequately addressed (e.g., ablating the order of the two stages in the instruction tuning step), causing a further increase in scores.

---

### Decision · Program_Chairs · 2025-01-22

Accept (Spotlight)